# The Bland-Altman method should not be used when one of the two measurement methods has negligible measurement errors

Patrick Taffé[ID]*, Claire Zuppinger[ID], Gerrit Marwin Burger[ID], Semira Gonseth Nusslé

Center for Primary Care and Public Health (Unisanté), University of Lausanne, Lausanne, Switzerland

* patrick.taffe@unisante.ch

**Data Availability Statement:** All relevant data are within the paper and its Supporting Information files.

## Abstract

### Background

The Bland-Altman limits of agreement (LoA) method is almost universally used to compare two measurement methods when the outcome is continuous, despite warnings regarding the often-violated strong underlying statistical assumptions. In settings where only a single measurement per individual has been performed and one of the two measurement methods is exempt (or almost) from any measurement error, the LoA method provides biased results, whereas this is not the case for linear regression.

### Methods

Thus, our goal is to explain why this happens and illustrate the advantage of linear regression in this particular setting. For our illustration, we used two data sets: a sample of simulated data, where the truth is known, and data from a validation study on the accuracy of a smartphone image-based dietary intake assessment tool.

### Results

Our results show that when one of the two measurement methods is exempt (or almost) from any measurement errors, the LoA method should not be used as it provides biased results. In contrast, linear regression of the differences on the precise method was unbiased.

### Conclusions

The LoA method should be abandoned in favor of linear regression when one of the two measurement methods is exempt (or almost) from measurement errors.

**Funding:** A research grant (Performance de l'outil digital de suivi de consommation alimentaire MyFoodRepo) was attributed to S. G. N. from the Committee for health promotion and fight against addictions of the Canton of Vaud (Commission de promotion de la santé et de lutte contre les addictions https://www.vd.ch/themes/sante-soins-et-handicap/prevention-et-maladies/appel-a-projets/exemples-de-projets-finances/). The funder had no role in the study design, data collection and analysis, decision to publish, or preparation of the manuscript.

**Competing interests:** The authors have declared that no competing interests exist.

# Introduction

Cross-validation of measurement methods, such as for diagnostic devices, ensures the accuracy of clinical data in innovation. The Bland-Altman's limits of agreement (LoA) method is one of the most widely applied statistical tools in medical research to assess the agreement/interchangeability of two measurement methods when the outcome is continuous (the 1986 paper, published in the Lancet journal [1], has been cited 51'779 times as of August 3, 2022, Google search). However, it has recently been shown to rely on strong statistical assumptions (1. The two measurement methods have the same precision, i.e. the measurement error variances are the same, 2. The precision is constant and does not depend on the true latent trait, i.e. the measurement error variances are constant, 3. The bias is constant, i.e. the difference between the two measurement methods is constant, aka there is only a differential bias) which, unfortunately, are often violated in practice [2–6]. Nevertheless, as exemplified by the numerous (and still in augmentation citations), many researchers seem to be unaware of its important limitations. Clearly, there is a need for challenging the relevance of using LoA method in different usual experimental settings, particularly when one of the two measurement methods is known to be exempt (or almost) from measurement errors.

Given the almost universal use of the LoA method, we found it important to investigate the use of the LoA method when one of the two measurement methods is known to be exempt (or almost) from measurement errors. This is typically the case when the device used to measure the true latent trait is very precise so that repeated measurements on the same individual would result in the same (or almost) value each time. To be concrete and motivate our presentation, consider the example of a dietician weighting the different food items on a plate with a very precise weighing scale and assessing the nutritional value of each item in terms of caloric intake [6]. If she were to repeat measurements, the same quantity (or almost) of each food item would be found and the same number of calories (or almost) would be assessed and measurement errors would be almost null or at least very small. Now, as employing a dietician is costly and may not be always feasible, imagine using, instead, a smartphone image-based dietary intake assessment tool, based on a special application allowing to recognize the photographed items on the plate and their quantity. This measuring device may be used on a large scale very easily but may not be as precise as the dietician. Actually, given the novelty of the technology and the difficulty to recognize the food items and assess their quantity based solely on a smartphone picture, it may be anticipated that measurement errors of this measuring device may be non-negligible. Before making the application available to the public, it is desirable to assess the agreement/interchangeability between the two measurement methods (i.e. dietician versus smartphone).

In this setting, it would be ill-advised to use the LoA method to compare the two measurement methods, as the required underlying statistical assumptions to validly use this methodology are violated [2, 4]. Indeed, it has been shown that regression of the differences (y1-y2) on the means (y1+y2)/2 (i.e. the LoA method) provides unbiased estimates of the bias (which can be decomposed into differential and proportional biases) only when the ratio of the variances of measurement errors is strictly proportional to the proportional bias (see equation (10) in Taffé [2]), a condition certainly violated when one of the two measurement methods is almost exempt from measurement errors.

When individual repeated measurements are available by at least one of the two measurement methods, Taffé [2, 7] has developed a new methodology to assess bias, precision, and agreement between the two measurement methods, which circumvents the deficiencies of the Bland-Altman LoA method. However, when individual repeated measurements are not available and there is only one measurement per individual, by each instrument, applied

researchers may still be tempted to use the LoA method, despite the strong and often violated underlying statistical assumptions, as it does not rely on individual repeated measurements.

The goal of this report is to show that when only one measurement per individual is available, by each instrument, and the reference method is almost exempt from measurement errors, the use of the LoA method is discouraged, as it provides biased estimates. It will be demonstrated that, when the signal-to-noise ratio is large (e.g. at least 100), in the sense that the amplitude of the true signal is much larger than measurement errors, or in statistical terms, the variance of the true latent trait is much larger than that of measurement errors, then simple regression analysis of the measurements by the new method (y1), or of the differences (y1-y2), on the reference method (y2) provides unbiased estimates.

## Material and methods

To answer this question, we have carried out a simulation study (so that the truth is known) and assessed the performance of both the conventional LoA method and, as an alternative statistical method, linear regression of the measurements by the device plagued by measurement errors (i.e. y1 as dependent variable) on the measurements by the device exempt of measurement errors (i.e. y2 as the independent variable), as well as regression of the differences (y1-y2) on the measurements by the device exempt of measurement errors (y2). To provide a concrete example, using both methods we have analyzed data from a validation study on the accuracy of a smartphone image-based dietary intake assessment tool in terms of caloric content compared with the evaluation made by a dietician [6].

It is useful to recall that with the LoA method one of the two measurement methods, say y2, is implicitly taken as the reference and the other, y1, is compared to it by computing the average of the differences (e.g. y1-y2) to estimate the average/differential bias. The sign of the bias will depend on which difference is computed, either y1-y2 or y2-y1, and consequently on the method used as the reference (note that, here, "reference" means the benchmark for the comparison and not that the method deemed as "reference" is unbiased or without measurement error) [4].

For the simulation study, data have been generated by considering that measurement method2 (y2) is unbiased and has almost no measurement errors (i.e. the variance of the measurement errors is set to a very small level, but not to zero as it is fundamentally almost impossible to measure anything without measurement errors):

$$\text{measurement method2} = \text{true trait} + \text{measurement error2}$$

which may be formally and compactly written:

$$y2 = x + \text{error2}$$

$$\text{error2} \sim \text{N}(0, \ \sigma^2_{error2})$$

where the true trait $x$ takes values between 25 and 50, according to a uniform distribution with mean $\mu_x = 37.5$ and variance $\sigma^2_x = 52.08$ (see Fig 1 below, where the x-axis represents the true latent trait and the y-axis the measurements made by the two measurement methods, which clearly suffer from measurements errors as the points are not all aligned on the 45˚ line). To mimic real-world conditions, despite being very small, the variance $\sigma^2_{error2}$ of measurement error2 is assumed to be increasing with the level of the true trait $x$:

$$\sigma^2_{error2} = (0.01 + 0.01x)^2$$

For example, when $x = 25$, $\sigma^2_{error2} = 0.0676$, and when $x = 50$, $\sigma^2_{error2} = 0.2601$. Clearly, even when the true trait $x$ takes value 50, the variance of the measurement errors $\sigma^2_{error2} = 0.2601$ is much smaller than the variance of the true trait $\sigma^2_x = 52.08$ and the signal-to-noise ratio $\sigma^2_x / \sigma^2_{error2}$ is about 200, which means that measurement errors remain negligible.

Assuming that measurement errors are increasing with the level of the true trait is a rather natural assumption, which can be observed in many practical experiments. For example, in a study on energy expenditure in ventilated critically ill children, we found that the variance of measurement errors was increasing with the level of energy expenditure [5]. In another study on the bias and precision of oscillometric devices, we also found that the variance of measurement errors was increasing with the level of blood pressure [3]. Finally, in a very recently published study, we found that the variance of measurement errors when measuring the energy content of a meal using a smartphone application was increasing with the true caloric content assessed by a dietician [6].

Regarding method1 (y1), it is assumed to be plagued by both a differential bias (i.e. a constant difference between the true trait and the measurement method) and a proportional bias (i.e. a difference which depends on the value of the true trait):

measurement method1 = differential bias + proportional bias * true trait + measurement errors1

which may be formally and compactly written:

$$y1 = \alpha + \beta * x + \text{error1}$$

Consequently, the (total) bias (i.e. the systematic difference $E(y1 - x)$ between $y1$ and $x$) is given by:

$$\text{bias} = \alpha + x*(\beta - 1)$$

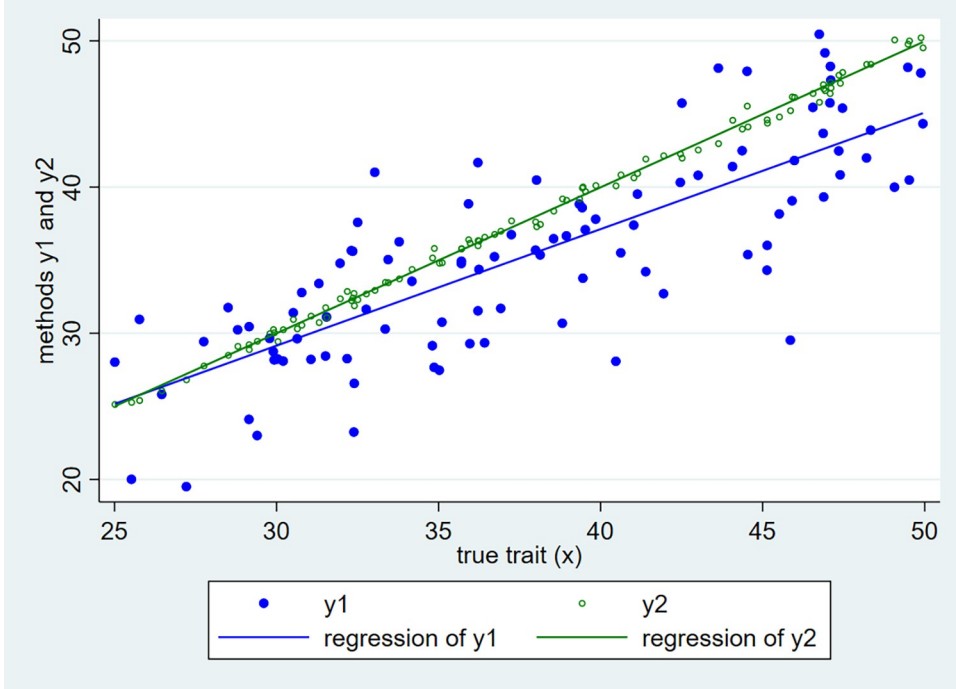

**Fig 1. Simulated data.**

This formulation makes it clear that the (total) bias depends on the differential bias (α), which represents the constant difference between the true trait and the measurement method, whatever the value $x$ of the true trait, and the proportional bias (β ≠ 1), which takes into account that the amount of bias may depend on the value of the true trait $x$. It is only when α = 0 and β = 1 that the measurement method is unbiased.

For our simulation, the differential bias has been set to 7 (i.e. α = 7) and the proportional bias to 0.75 (i.e. β = 0.75). Therefore, in our simulated data the amount of bias is 0.75 when $x$ = 25 and -5.5 when $x$ = 50.

Contrary to method2 (y2), we assumed that method1 (y1) bears a non-negligible amount of measurement errors given by:

$$\sigma^2_{error1} = (0.01 + 0.01x)^2 + 23$$

For example, when $x$ = 25, $\sigma^2_{error1}$ = 23.0676, and when $x$ = 50, $\sigma^2_{error1}$ = 23.2601. When the true trait $x$ is 50 the signal-to-noise ratio $\sigma^2_x / \sigma^2_{error1}$ is about 2.2, which represents a setting where measurement errors are large.

The simulated data are represented in Fig 1 below:

Clearly, with respect to method2 (y2), method1 (y1) exhibits a non-constant negative bias (the blue regression line lies below the green one), which depends on the value of the true trait $x$: the bias is quasi null around 25 and about -5 when the true trait is 50. Also, method1 is much less precise than method2: the blue points are much more dispersed than the green ones. Note, though, that method2 exhibits a small amount of measurement errors (the green points do not lie exactly on the green regression line, there is some scatter), as expected in real-world settings.

For the readership not acquainted with the Bland-Altman's LoA method, recall that in the conventional LoA methods the differences (y1-y2) (i.e. the dependent variable represented in the y-axis), are regressed on the means (y1+y2)/2 (i.e. the independent variable represented in the x-axis) and the average bias is simply estimated by the mean of the differences [1]. A regression line is sometimes superimposed on the graph to allow for the presence of a proportional bias in addition to the differential bias [8].

## Results

Now, we are ready to illustrate what happens when using the LoA method to assess the agreement/interchangeability between the two measurement methods:

On Fig 2, the conventional LoA method indicates a differential bias of -2.2 (i.e. α = -2.2, 95%CI[-7.4;3.0]), and no proportional bias (i.e. β = 0.99, 95%CI[0.86;1.13]) (see ref. [4] for details regarding the computation of the differential and proportional biases when using the LoA method). Notice also that the regression line of the differences on the averages is almost confounded with the observed average agreement (i.e. the mean of the differences), as there is no proportional bias. As the data have been simulated, we know that there is a differential bias of 7 and a proportional bias of 0.75. Clearly, in this example, the LoA method is misleading, as the true bias is null for values of the true trait near 25 and about -5 when the true trait is 50 (see Fig 1).

Turning to the linear regression of the measurements by the device plagued by measurement errors (y1) on the measurements by the device exempt or almost of measurement errors (y2), one gets:

Fig 3 represents a scatter plot of measurements y1 versus y2. The black regression line is the 45˚ reference line of no bias, whereas the green line represents the regression of y1 on y2 (i.e. y1 = a + b*y2). When y2 is exempt from measurement errors coefficient "a" estimates the

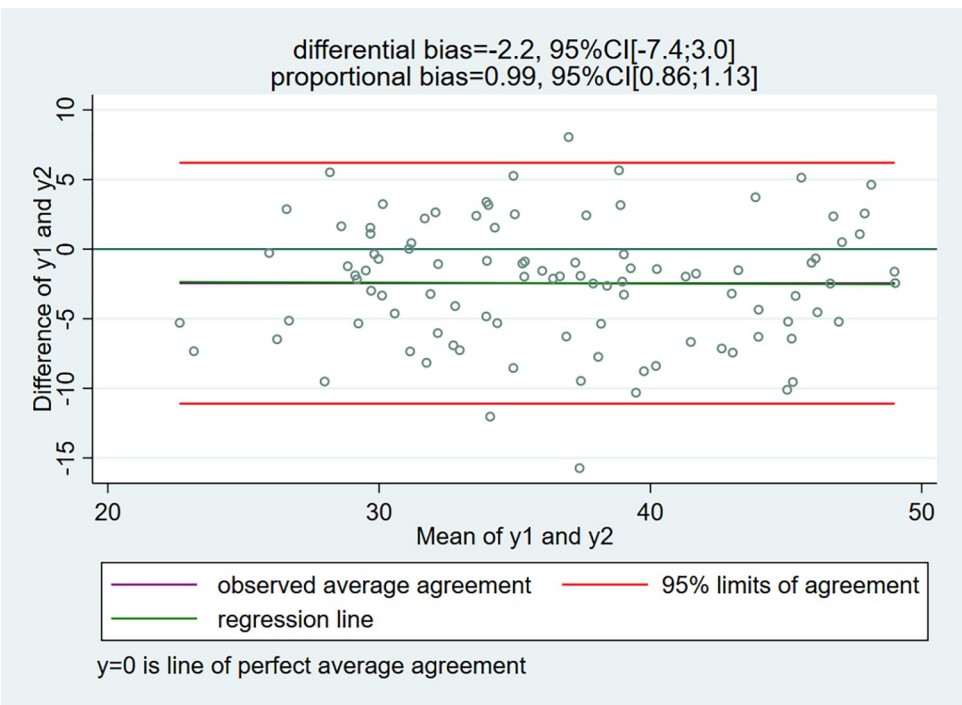

**Fig 2. Simulated data: Classical Bland-Altman LoA plot augmented with the regression line.**

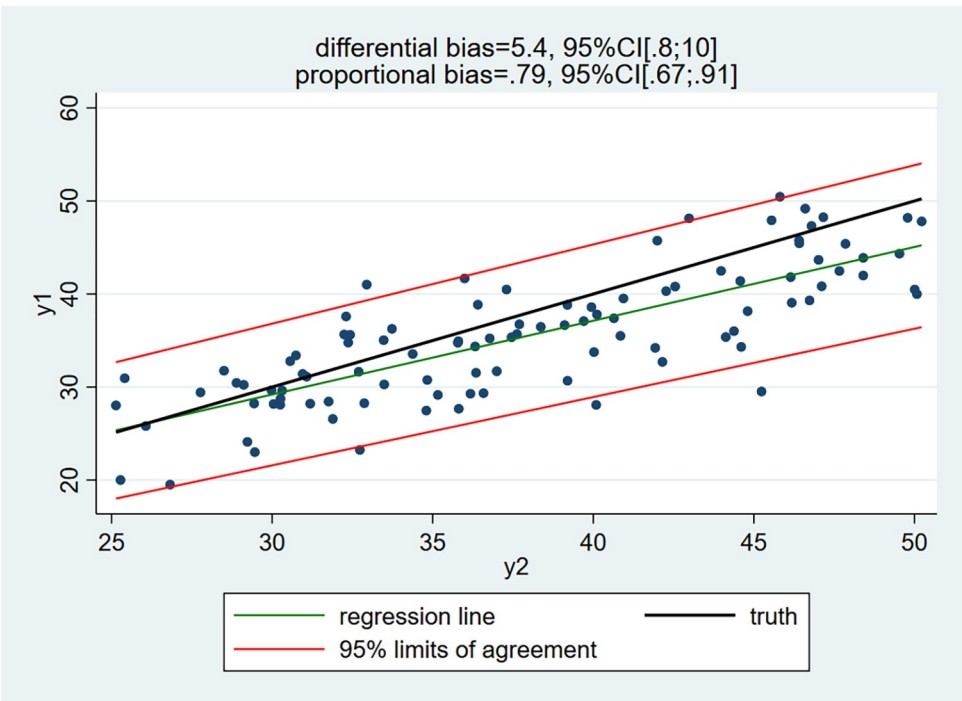

**Fig 3. Simulated data: Linear regression of y1 on y2 with 95% limits of agreement.**

differential bias and coefficient "b" the proportional bias. We obtained a = 5.4, 95%CI [0.8;10.0], and b = 0.79, 95%CI[0.67;0.91]. These intervals contain the true values of the differential = 7 and proportional = 0.75 biases. The green regression line indicates that method1 (y1) is unbiased for values of y2 (method2) near 25 and biased for larger values; the bias increases gradually and amounts to about 5 when y2 = 50. Clearly, in this example, the regression method provides unbiased estimates, whereas the Bland-Altman method is biased.

The 95% limits of agreement exhibit a slight funnel-shaped pattern, as the variance of the measurement errors is (slightly) increasing with the value of the true trait. The limits are computed by a methodology based on the absolute value of the residuals [2].

To provide a similar figure as the traditional LoA plot, we have also considered the regression of the differences (y1 –y2) on the measurements by the device exempt (or almost) of measurement errors (y2):

Fig 4 resembles more the standard LoA plot except that the x-axis is not the mean (y1 + y2)/ 2 but y2. As method2 (y2) is exempt (or almost) of measurement errors, the coefficients of the regression of the differences (y1 –y2) on y2 (i.e. (y1 –y2) = a + b $^*$y2) allows one to estimate the differential = a and proportional = b + 1 biases. One gets exactly the same values as above with the regression y1 on y2. Consequently, the regression of the differences (y1 –y2) on y2 provides an alternative valid method to estimate the bias of method1 (with respect to method2).

The green regression line of the differences (y1 –y2) on y2 clearly illustrates that method1 (y1) is unbiased for values of y2 (method2) near 25 and biased for larger values; the bias increases gradually and amounts to about 5 when y2 = 50. Again, it is clear in this example that the regression method provides unbiased estimates, whereas the Bland-Altman method is biased.

These results are easily confirmed by carrying out simulation analyses. However, analytic results (when possible) are more general and we formally explain below why the Bland-Altman LoA method is biased, whereas linear regression of the measurements by the device plagued by measurement errors (y1 as the dependent variable) or of the differences (y1-y2 as the

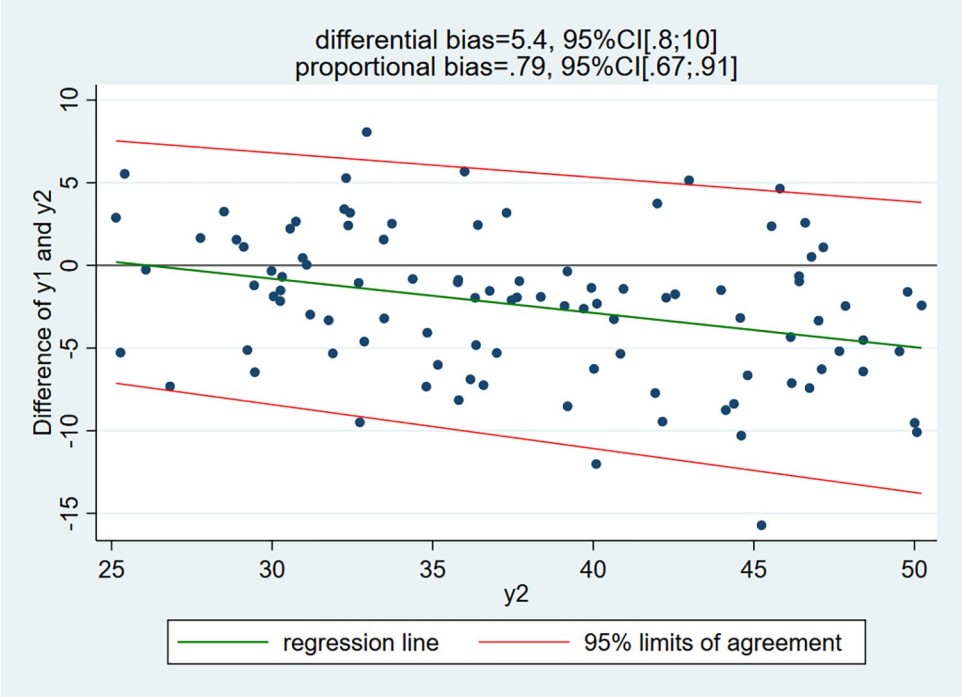

**Fig 4. Simulated data: Linear regression of the differences (y1—y2) on y2 with 95% limits of agreement.**

dependent variable) on the measurements by the device exempt of measurement errors (y2 as the independent variable) provides unbiased estimates.

Why the Bland-Altman LoA method is biased?

Actually, an important assumption of the linear regression model, that is sometimes over-looked, is that regressors are measured without measurement errors. This assumption is violated in the Bland-Altman method when regressing the differences on the means, whereas this is not the case when using y2 instead of the mean, as by assumption y2 is exempt (or almost) of measurement errors.

One can show that in the linear regression model y1 = α + β * y2 + error, where the independent variable y2 is measured with errors (i.e. y2 = $x$ + error2), the estimate b (of β) will converge to $\beta/(1 + \sigma^2_{error2}/\sigma^2_x)$ when the sample size increases indefinitely. This formula shows that when the variance $\sigma^2_{error2}$ of the measurement errors is small, with respect to the variance $\sigma^2_x$ of the true latent trait $x$, then $(1 + \sigma^2_{error2}/\sigma^2_x)$ is almost equal to 1 and the estimate b is almost equal to the true parameter β. The term $\sigma^2_{error2}/\sigma^2_x$ is exactly the inverse of the signal-to-noise ratio $\sigma^2_x/\sigma^2_{error2}$, which shows that when the signal-to-noise ratio is large, regression estimates are almost unbiased.

Above, we suggested a signal-to-noise ratio of at least 100 or more to get very accurate estimates. Notice, however, that without repeated measurements from method2 (y2) it is not possible to identify the two variances $\sigma^2_x$ and $\sigma^2_{error2}$, and one must rely on outside information (e.g. other studies) to appraise the likely value of the signal-to-noise ratio.

## Illustration based on a concrete example

To illustrate the advantages of the proposed regression approach (either the regression of y1 on y2 or of the differences (y1-y2) on y2) compared to the commonly used Bland-Altman method, we have used both methods to analyze data from a validation study on the accuracy of a smartphone image-based dietary intake assessment tool to quantify the caloric content of different meals [6]. Here, the measurements made by the smartphone application are labeled kcalMFR (y1) and those computed by the dietician kcalIU (y2, the reference method). We assume that measurements made by the dietician are (almost) exempt from measurement errors, and were she to reassess the weight and caloric content of the different food items from the meals she would obtain (almost) exactly the same results (as she used a very precise weighing scale up to the gram).

As we have shown above, the regression of y1 on y2 or of the differences (y1-y2) on y2 provides exactly the same estimates and we focus, here, on the latter:

Based on the Bland-Altman method (Fig 5 left) one may believe that the smartphone application provides an almost unbiased estimate of the caloric content of the meals (the green regression line is almost flat on zero), whereas the regression method indicates that the application over-estimates the caloric content of low-calorie meals (less than 200 kcal) but under-estimates the caloric content of high-calorie meals (larger than 200 kcal).

However, the inspection of the plots illustrates that there are two clusters of meals, one whose food items present a caloric content below 200 kcal and the other with higher calorie food items. Therefore, we have repeated the analyses separately for the two groups:

In Fig 6 (left), the Bland-Altman method seems to indicate (green regression line) that the smartphone application underestimates the caloric content up to about 60 kcal and over-estimates afterward. The regression method (right) tells a different story, the smartphone application overestimates the caloric content, particularly for low-calorie food items, and is more accurate when the caloric content is about 200 kcal.

Turning to the second group of food items:

In Fig 7 (left), the Bland-Altman method seems to indicate (green regression line) that the smartphone application underestimates the caloric content of food items containing between

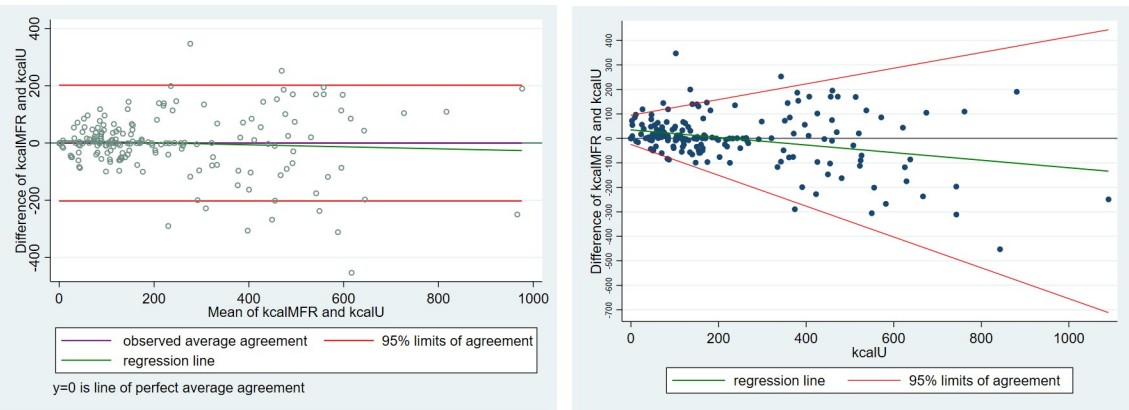

**Fig 5.** (Left) Classical Bland-Altman LoA plot augmented with the regression line (right) Linear regression of the differences (y1—y2) on y2 with 95% limits of agreement.

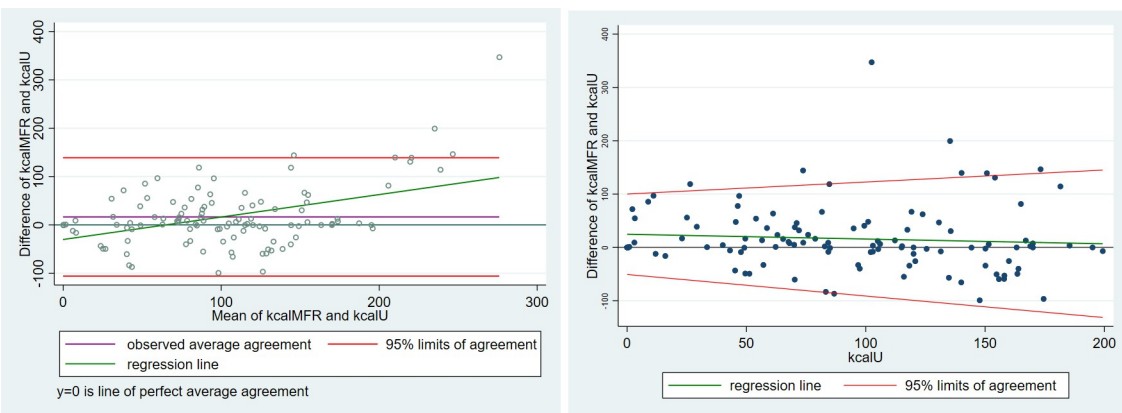

**Fig 6.** (Left) Classical Bland-Altman LoA plot augmented with the regression line, for low-caloric aliment cluster (right) Linear regression of the differences (y1—y2) on y2 with 95% limits of agreement.

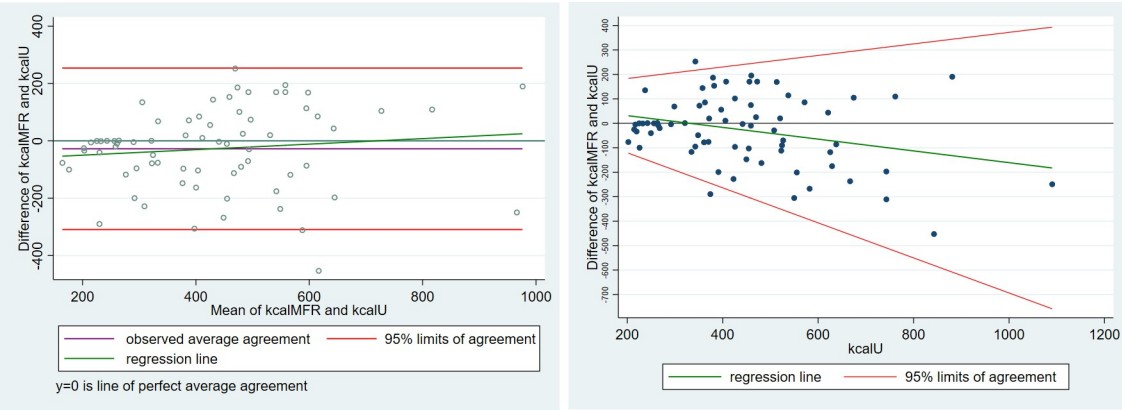

**Fig 7.** (Left) Classical Bland-Altman LoA plot augmented with the regression line, for high-caloric aliment cluster (right) Linear regression of the differences (y1—y2) on y2 with 95% limits of agreement.

200 and 700 kcal, and over-estimates afterward. However, the regression method (right) shows that the smartphone application increasingly underestimates the caloric content of food items containing more than 400 kcal.

## Take-home message

- When one of the two measurement methods is exempt (or almost) of measurement errors, the Bland-Altman method should not be used and regression of the differences (y1-y2) on the precise method y2 or of y1 on y2 should be preferred.

- When the two measurement methods are known to have non-negligible measurement errors (in the sense that the signal-to-noise ratio is much lower than 100), the Bland and Altman method can be used as long as the following statistical assumptions do hold [2, 4]: 1) absence of any proportional bias and 2) equal and 3) constant variances of the measurement errors.

- When these three assumptions do not hold, which is often the case in practice, one must gather repeated measurements from at least one of the two measurement methods and use a more complex existing statistical methodology to analyze the data [2, 7, 9, 10]. The methodology developed in references [2] and [7] has been made available in the Stata [11, 12] and (some part of it) R packages [13]. The methodology developed in this report will be made available in a future Stata package.

## Discussion

In this paper, we have stuck to the original Bland-Altman LoA method where a single measurement per individual is available by each measurement method [1]. However, the methodology has been extended by these authors to the setting of repeated measurements per individual [8] (we will refer to it as the "extended Bland-Altman LoA method"), but this is not the topic of this report as the focus is explicitly on the setting where only a single measurement per individual is available and one of the two measurement methods is exempt or almost of measurement errors.

The Bland-Altman LoA method is simple to use and understand. However, it has been shown to rely on strong statistical assumptions, which, unfortunately, are often violated in practice [2–6]. When individual repeated measurements are available by at least one of the two measurement methods, various statistical methods have been developed to circumvent the deficiencies of the extended Bland-Altman LoA method [2, 7, 9, 10]. However, these methodologies are not applicable with a single measurement per individual and an investigator may be tempted to use the (conventional) Bland-Altman LoA method despite violation of the underlying statistical assumptions (e.g. in settings where one of the two measurement methods is exempt or almost of measurement errors). We have shown that in this case, it would be ill-advised to use the Bland-Altman method and simple linear regression analysis of the differences or the measurement method plagued by measurement error on the precise method provides unbiased estimates.

There is a debate in the literature regarding the pro and cons of the Bland-Altman LoA method [14–17]. Unfortunately, in these papers, the genuine reason for the bias of the method, when the underlying statistical assumptions are violated, has not been identified as being a problem of endogeneity, which has somewhat obscured the debate and it is only very recently that this problem has been clarified and solved [2, 7, 9, 10]. It requires, however, repeated measurements per individual, otherwise, it is not possible to disentangle the differential from the proportional bias.

## Conclusions

We have highlighted that when one of the two measurement methods is exempt (or almost) from measurement errors, the Bland-Altman method should not be used. In this setting, regression of the differences (y1-y2) on the precise method y2 or of y1 on y2 should be preferred. The choice of the statistical methods has thus important implications for the validity of studies using cross-validation of measurement methods.

## Supporting information

**S1 Data. This file contains the simulated data.**
(XLSX)

**S2 Data. This file contains the data used in "Illustration based on a concrete example".**
(XLSX)

**S1 File.**
(TXT)

**S2 File.**
(TXT)

## Author Contributions

**Conceptualization:** Patrick Taffé, Claire Zuppinger, Gerrit Marwin Burger, Semira Gonseth Nusslé.

**Formal analysis:** Patrick Taffé.

**Methodology:** Patrick Taffé, Claire Zuppinger, Gerrit Marwin Burger, Semira Gonseth Nusslé.

**Writing – original draft:** Patrick Taffé, Claire Zuppinger, Gerrit Marwin Burger, Semira Gonseth Nusslé.

**Writing – review & editing:** Patrick Taffé, Claire Zuppinger, Gerrit Marwin Burger, Semira Gonseth Nusslé.

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
