## [Decision Letter · Decision Letter 0]

27 Jul 2022

PONE-D-22-16067The Bland-Altman method should not be used when one of the two measurement methods has negligible measurement errorsPLOS ONE

Dear Dr. Taffé,

Thank you for submitting your manuscript to PLOS ONE. After careful consideration, we feel that it has merit but does not fully meet PLOS ONE’s publication criteria as it currently stands. Therefore, we invite you to submit a revised version of the manuscript that addresses the points raised during the review process.

ACADEMIC EDITOR:I have read this paper multiple times and it is indeed very interesting. Together with the suggestions from Reviewer 1 i would also suggest the following changes/additions:- Please provide your Data generating mechanisms/code as supplemental material. - Line 47 - Please update the number of citations and reference to google scholar, instead os just a google search (27 July 2022 - 51,759).- Line 49 - Please inform the reader of the actual assumptions.

We look forward to receiving your revised manuscript.

Kind regards,

Markus Harboe Olsen

Academic Editor

PLOS ONE

Journal Requirements:

2. Please note that PLOS journals require authors to make all data necessary to replicate their study’s findings publicly available without restriction at the time of publication. Please see our Data Availability policy at https://journals.plos.org/plosone/s/data-availability. As such, please make the full specific datasets used in this study available by either A) uploading the full datasets as supplementary information files, or B) including a URL link in your Data Availability Statement and Methods section to where the full datasets can be accessed.

Reviewers' comments:

Reviewer's Responses to Questions

**Comments to the Author**

1. Is the manuscript technically sound, and do the data support the conclusions?

Reviewer #1: Yes

Reviewer #2: Yes

2. Has the statistical analysis been performed appropriately and rigorously? 

Reviewer #1: Yes

Reviewer #2: Yes

3. Have the authors made all data underlying the findings in their manuscript fully available?

Reviewer #1: No

Reviewer #2: No

4. Is the manuscript presented in an intelligible fashion and written in standard English?

Reviewer #1: Yes

Reviewer #2: Yes

5. Review Comments to the Author

Reviewer #1: A generally well written manuscript in good English. I detected no obvious o=typographical or grammatical errors.

There are some points to consider.

Introduction. This sets the scene well, however, at line 49 and then from line 75 onward, the authors refer to violating the underlying assumptions of the B&A method and refer to three of their prior publications. It would be extremely helpful to state here exactly what these assumptions are. This is important since it underscores the need for this paper. It is remiss to expect the reader to have to seek out these other publications particularly since I assume the hope of the authors is to alert the statistically naïve reader to the fallibilities of the B&A method.

Line 112. I am not convinced that it is always the case or implicit that one method is assumed as the reference. The original B&A approach was not designed with this in mind but simply to compare different methods for interchangeability not how well one performs against a standard method. In this case the x axis is the mean of methods. Later modifications of the B&A approach have certainly assumed a reference method and X axis values become those of the method. See DOI: 10.1002/sim.3086. Please justify or modify your statements.

Lines 130-140. The values provided here are presumably for the magnitude of the data set being used but this is not clear. Please provide a few details of the dataset (or provide as supplementary data). Again for the naïve reader the source of these numeric values will be obscure.

Line 149-150. It would be useful here to define formally for the reader what is meant by "differential" and "proportional" biases rather than inferring this from the previous sentences or waiting until lines around line 167 to find out.

Line 195. My point above - the mean of methods is not necessarily used when one assumes a reference method (or one with minimal error).

The Results are generally clearly presented.

The Discussion is a misnomer. There is no Discussion it is simply a list of key findings. The authors need to place their observations and criticisms of the B&A method in context of papers which are pro the B&A approach (e.g. https://doi.org/10.1016/j.gloepi.2020.100045) and those that are negative (e.g. Sportscience, 8 (2004), pp. 42-46 and Krouwer cited above).

In addition, while noting the need to account for variation in measurements (error) they have omitted to mention the B and A modified their original method to take this into account (see DOI: 10.1177/096228029900800204 and has been discussed elsewhere https://doi.org/10.1093/bja/aem214) This latter paper recognizes that the standard B&A cannot be applied where repeated measures are available, i.e. the error is known and albeit potentially is negligible as in the authors examples. Discussion on these points is relevant and would be useful.

Reviewer #2: It is a very well written paper. Bland-Altman plot is an excellent analysis tool for evaluating agreement between instruments. The authors have very clearly shown its shortcomings and have also suggested a new method for a more insightful analysis. A minor suggestion is that the 95% confidence intervals of differential and proportional biases should be added to the bias plot.

6. PLOS authors have the option to publish the peer review history of their article (what does this mean?). If published, this will include your full peer review and any attached files.

Reviewer #1: No

Reviewer #2: No

---

## [Author Response · Author response to Decision Letter 0]

5 Aug 2022

PONE-D-22-16067

The Bland-Altman method should not be used when one of the two measurement methods has negligible measurement errors

PLOS ONE

Dear Dr. Taffé,

Thank you for submitting your manuscript to PLOS ONE. After careful consideration, we feel that it has merit but does not fully meet PLOS ONE’s publication criteria as it currently stands. Therefore, we invite you to submit a revised version of the manuscript that addresses the points raised during the review process.

ACADEMIC EDITOR:

I have read this paper multiple times and it is indeed very interesting. 

Thank you for the nice comment.

Together with the suggestions from Reviewer 1 i would also suggest the following changes/additions:

- Please provide your Data generating mechanisms/code as supplemental material. 

Data have been made available by uploading them as supplementary information files.

- Line 47 - Please update the number of citations and reference to google scholar, instead os just a google search (27 July 2022 - 51,759).

Done.

- Line 49 - Please inform the reader of the actual assumptions.

Done.

The sentence now reads:

However, it has recently been shown to rely on strong statistical assumptions (1. The two measurement methods have the same precision, i.e. the measurement error variances are the same, 2. The precision is constant and does not depend on the true latent trait, i.e. the measurement error variances are constant, 3. The bias is constant, i.e. the difference between the two measurement methods is constant, aka there is only a differential bias), which, unfortunately, are often violated in practice [2-4].

We look forward to receiving your revised manuscript.

Kind regards,

Markus Harboe Olsen

Academic Editor

PLOS ONE

Journal Requirements:

Done.

2. Please note that PLOS journals require authors to make all data necessary to replicate their study’s findings publicly available without restriction at the time of publication. Please see our Data Availability policy at https://journals.plos.org/plosone/s/data-availability. As such, please make the full specific datasets used in this study available by either A) uploading the full datasets as supplementary information files, or B) including a URL link in your Data Availability Statement and Methods section to where the full datasets can be accessed.

Data have been made available by uploading them as supplementary information files.

There is no grant number as S. G. N. received a reseach grant by the Commission de promotion de la santé et de lutte contre les addictions (CPSLA), in english Committee for health promotion and fight against addictions of the Canton of Vaud, which is a public institution from the canton of Vaud, Switzerland. We have added the URL of CPSLA.

As mentioned above, the data have been made available.

Done.

Reviewers' comments:

Reviewer's Responses to Questions

Comments to the Author

1. Is the manuscript technically sound, and do the data support the conclusions?

Reviewer #1: Yes

Reviewer #2: Yes

2. Has the statistical analysis been performed appropriately and rigorously?

Reviewer #1: Yes

Reviewer #2: Yes

3. Have the authors made all data underlying the findings in their manuscript fully available?

Reviewer #1: No

Reviewer #2: No

This has been corrected and the data have been made available by uploading them as supplementary information files.

4. Is the manuscript presented in an intelligible fashion and written in standard English?

Reviewer #1: Yes

Reviewer #2: Yes

5. Review Comments to the Author

Reviewer #1: A generally well written manuscript in good English. I detected no obvious o=typographical or grammatical errors.

Thank you for the nice comment.

There are some points to consider.

Introduction. This sets the scene well, however, at line 49 and then from line 75 onward, the authors refer to violating the underlying assumptions of the B&A method and refer to three of their prior publications. It would be extremely helpful to state here exactly what these assumptions are. This is important since it underscores the need for this paper. It is remiss to expect the reader to have to seek out these other publications particularly since I assume the hope of the authors is to alert the statistically naïve reader to the fallibilities of the B&A method.

Done.

The sentence now reads:

However, it has recently been shown to rely on strong statistical assumptions (1. The two measurement methods have the same precision, i.e. the measurement error variances are the same, 2. The precision is constant and does not depend on the true latent trait, i.e. the measurement error variances are constant, 3. The bias is constant, i.e. the difference between the two measurement methods is constant, aka there is only a differential bias), which, unfortunately, are often violated in practice [2-4].

Line 112. I am not convinced that it is always the case or implicit that one method is assumed as the reference. The original B&A approach was not designed with this in mind but simply to compare different methods for interchangeability not how well one performs against a standard method. In this case the x axis is the mean of methods. Later modifications of the B&A approach have certainly assumed a reference method and X axis values become those of the method. See DOI: 10.1002/sim.3086. Please justify or modify your statements.

Actually, your remark entails several questions we will answer.

First, when computing the differences and taking the mean to estimate the amount of differential bias you implicitly make a choice regarding the method taken as reference, as the sign of the bias will depend on it. Here, “reference” does not mean an unbiased method or one with no measurement errors; it is just defined by the contrast considered, either y1-y2, in which case y2 is the reference, or y2-y1, in which case y1 is the reference.

Second, whether the x-axis represents the mean of the two measurements or only one of the two methods is another question. Bland and Altman (Lancet 1995;346:1085-1087) recommend the average, whereas Krouwer (Stat in Med 2008;27.778-780) suggests the “reference” method. What the latter author calls the “reference” method is unclear as he does not define it and simply mentions that Bland and Altman call it “gold standard”. To our best understanding “gold standard” meant unbiased but did not imply being exempt from measurement errors. The point raised in this report is that the regression should use as the independent variable (i.e. x-axis) a measurement method exempt or almost of measurement errors. However, this independent variable need not be an unbiased measurement method, in which case the two measurement methods are simply compared to assess their exchangeability without reference to the truth. When the comparison implies the truth one requires repeated measurements and statistical methods which make explicit use of this truth such as those developed in references [2,7,9,10].

To address your remark we have reformulated the paragraph:

It is useful to recall that with the LoA method one of the two measurement methods, say y2, is implicitly taken as the reference and the other, y1, is compared to it by computing the average of the differences (e.g. y1-y2) to estimate the average/differential bias. The sign of the bias will depend on which difference is computed, either y1-y2 or y2-y1, and consequently on the method used as the reference (note that, here, “reference” means the benchmark for the comparison and not that the method deemed as “reference” is unbiased or without measurement error) [4].

Lines 130-140. The values provided here are presumably for the magnitude of the data set being used but this is not clear. Please provide a few details of the dataset (or provide as supplementary data). Again, for the naïve reader the source of these numeric values will be obscure.

For the sake of clarity, we have added the following sentence in parenthesis:

(see Figure 1 below, where the x-axis represents the true latent trait and the y-axis the measurements made by the two measurement methods, which, clearly, suffer from measurements errors as the points are not all aligned on the 45° line)

In addition, we have added a very recent reference [6] regarding the link between the measurement error variance and the true trait value:

Assuming that measurement errors are increasing with the level of the true trait is a rather natural assumption, which can be observed in many practical experiments. For example, in a study on energy expenditure in ventilated critically ill children, we found that the variance of measurement errors was increasing with the level of energy expenditure [5]. In another study on the bias and precision of oscillometric devices, we also found that the variance of measurement errors was increasing with the level of blood pressure [3]. Finally, in a very recently published study, we found that the variance of measurement errors, when measuring the energy content of a meal using a smartphone application, was increasing with the true caloric content assessed by a dietician [6].

Line 149-150. It would be useful here to define formally for the reader what is meant by "differential" and "proportional" biases rather than inferring this from the previous sentences or waiting until lines around line 167 to find out.

For the sake of clarity, we have reformulated the paragraph:

Regarding method1 (y1), it is assumed to be plagued by both a differential bias (i.e. a constant difference between the true trait and the measurement method) and a proportional bias (i.e. a difference which depends on the value of the true trait)

Line 195. My point above - the mean of methods is not necessarily used when one assumes a reference method (or one with minimal error).

See response to Line 112 above.

For the sake of clarity, we have added in parenthesis the reference to the y- and x-axis:

For the readership not acquainted with the Bland-Altman’s LoA method, recall that in the conventional LoA methods the differences (y1-y2) (i.e. the dependent variable represented in the y-axis), are regressed on the means (y1+y2)/2 (i.e. the independent variable represented in the x-axis) and the average bias is simply estimated by the mean of the differences [1]. A regression line is sometimes superimposed on the graph to allow for the presence of a proportional bias in addition to the differential bias [8].

The Results are generally clearly presented.

The Discussion is a misnomer. There is no Discussion it is simply a list of key findings. The authors need to place their observations and criticisms of the B&A method in context of papers which are pro the B&A approach (e.g. https://doi.org/10.1016/j.gloepi.2020.100045) and those that are negative (e.g. Sportscience, 8 (2004), pp. 42-46 and Krouwer cited above). In addition, while noting the need to account for variation in measurements (error) they have omitted to mention the B and A modified their original method to take this into account (see DOI: 10.1177/096228029900800204 and has been discussed elsewhere https://doi.org/10.1093/bja/aem214) This latter paper recognizes that the standard B&A cannot be applied where repeated measures are available, i.e. the error is known and albeit potentially is negligible as in the authors examples. Discussion on these points is relevant and would be useful.

Yes, we completely agree.

We have added a new discussion paragraph, as well as additional references:

Discussion

In this paper, we have stuck to the original Bland-Altman LoA method where a single measurement per individual is available by each measurement method [1]. However, the methodology has been extended by these authors to the setting of repeated measurements per individual [8] (we will refer to it as the “extended Bland-Altman LoA method”), but this is not the topic of this report as the focus is explicitly on the setting where only a single measurement per individual is available and one of the two measurement methods is exempt or almost of measurement errors.

The Bland-Altman LoA method is simple to use and understand. However, it has been shown to rely on strong statistical assumptions, which, unfortunately, are often violated in practice [2-6]. When individual repeated measurements are available by at least one of the two measurement methods, various statistical methods have been developed to circumvent the deficiencies of the extended Bland-Altman LoA method [2,7,9,10]. However, these methodologies are not applicable with a single measurement per individual and an investigator may be tempted to use the (conventional) Bland-Altman LoA method despite violation of the underlying statistical assumptions (e.g. in settings where one of the two measurement methods is exempt or almost of measurement errors). We have shown that in this case, it would be ill-advised to use the Bland-Altman method and simple linear regression analysis of the differences or the measurement method plagued by measurement error on the precise method provides unbiased estimates.

There is a debate in the literature regarding the pro and cons of the Bland-Altman LoA method [14-17]. Unfortunately, in these papers, the genuine reason for the bias of the method, when the underlying statistical assumptions are violated, has not been identified as being a problem of endogeneity, which has somewhat obscured the debate and it is only very recently that this problem has been clarified and solved [2,7,9,10]. It requires, however, repeated measurements per individual, otherwise, it is not possible to disentangle the differential from the proportional bias.

See References for the additional references.

Reviewer #2: It is a very well written paper. Bland-Altman plot is an excellent analysis tool for evaluating agreement between instruments. The authors have very clearly shown its shortcomings and have also suggested a new method for a more insightful analysis. A minor suggestion is that the 95% confidence intervals of differential and proportional biases should be added to the bias plot.

Thank you for the nice comment.

We have added the estimated differential and proportional biases, as well as their 95% confidence intervals to the figures regarding the simulated data where the truth is known. However, we have not added these pieces of information regarding the concrete example based on the smartphone application, as in Figures 5 to 7 the focus is on the estimated regression line and no figures regarding the differential and proportional biases are provided (in addition it would reduce the space on the graph used to plot the data and make it almost unreadable).

6. PLOS authors have the option to publish the peer review history of their article (what does this mean?). If published, this will include your full peer review and any attached files.

Do you want your identity to be public for this peer review? For information about this choice, including consent withdrawal, please see our Privacy Policy.

Reviewer #1: No

Reviewer #2: No

---

## [Decision Letter · Decision Letter 1]

28 Nov 2022

The Bland-Altman method should not be used when one of the two measurement methods has negligible measurement errors

PONE-D-22-16067R1

Dear Dr. Taffé,

We’re pleased to inform you that your manuscript has been judged scientifically suitable for publication and will be formally accepted for publication once it meets all outstanding technical requirements.

Kind regards,

Markus Harboe Olsen

Academic Editor

PLOS ONE

Additional Editor Comments (optional):

Dear Authors, 

Your comments to the reviewers suggestions are acceptable. 

Reviewers' comments:

Reviewer's Responses to Questions

**Comments to the Author**

1. If the authors have adequately addressed your comments raised in a previous round of review and you feel that this manuscript is now acceptable for publication, you may indicate that here to bypass the “Comments to the Author” section, enter your conflict of interest statement in the “Confidential to Editor” section, and submit your "Accept" recommendation.

Reviewer #1: All comments have been addressed

2. Is the manuscript technically sound, and do the data support the conclusions?

Reviewer #1: Yes

3. Has the statistical analysis been performed appropriately and rigorously? 

Reviewer #1: Yes

4. Have the authors made all data underlying the findings in their manuscript fully available?

Reviewer #1: Yes

5. Is the manuscript presented in an intelligible fashion and written in standard English?

Reviewer #1: Yes

6. Review Comments to the Author

Reviewer #1: The authors have satisfactorily addressed the issues raised in my original review and I now support publication.

7. PLOS authors have the option to publish the peer review history of their article (what does this mean?). If published, this will include your full peer review and any attached files.

Reviewer #1: No

---

## [Editor Report · Acceptance letter]

1 Dec 2022

PONE-D-22-16067R1 

The Bland-Altman method should not be used when one of the two measurement methods has negligible measurement errors 

Dear Dr. Taffé:

I'm pleased to inform you that your manuscript has been deemed suitable for publication in PLOS ONE. Congratulations! Your manuscript is now with our production department. 

Kind regards, 

on behalf of

Dr. Markus Harboe Olsen 

Academic Editor

PLOS ONE